biomechanics

captivity, wild animals, bone density, trabecular bone, felids, mobility

**Author for correspondence:**
Habiba Chirchir
e-mail: chirchir@marshall.edu

# Effects of reduced mobility on trabecular bone density in captive big cats

Habiba Chirchir[1,2], Christopher Ruff[3], Kristofer M. Helgen[4] and Richard Potts[2]

[1]Marshall University, Huntington, WV 25755-0003, USA
[2]Human Origins Program, National Museum of Natural History, Smithsonian Institution, Washington, DC, USA
[3]Functional Anatomy and Evolution, Johns Hopkins University, Baltimore, MD 21205, USA
[4]Australian Museum Research Institute, Sydney 2010, Australia

 HC, 0000-0002-2400-3914; CR, 0000-0002-2932-3634

Bone responds to elevated mechanical loading by increasing in mass and density. Therefore, wild animals should exhibit greater skeletal mass and density than captive conspecifics. This expectation is pertinent to testing bone functional adaptation theories and to comparative studies, which commonly use skeletal remains that combine zoo and wild-caught specimens. Conservationists are also interested in the effects of captivity on bone morphology as it may influence rewilding success. We compared trabecular bone volume fraction (BVF) between wild and captive mountain lions, cheetahs, leopards and jaguars. We found significantly greater BVF in wild than in captive felids. Effects of captivity were more marked in the humerus than in the femur. A ratio of humeral/femoral BVF was also lower in captive animals and showed a positive relationship to home range size in wild animals. Results are consistent with greater forelimb than hindlimb loading during terrestrial travel, and possibly reduced loading of the forelimb associated with lack of predatory behaviour in captive animals. Thus, captivity among felids has general effects on BVF in the postcranial skeleton and location-specific effects related to limb use. Caution should be exercised when identifying skeletal specimens for use in comparative studies and when rearing animals for conservation purposes.

## 1. Introduction

Evidence accumulated over the past 100 years comparing organisms in captivity and the wild suggests a range of morphological changes owing to captivity [1–11]. Some of the more extreme changes observed in captive environments have

been skeletal and dental pathologies, attributed to poor diets and extremely limited mobility in unnatural settings [8,12–16]. Differences have also been observed in life-history parameters such as body size and growth rate. Captive animals generally grow bigger and mature faster compared to their wild counterparts ([6,17–20]; but see [21–23]). The larger cranial dimensions of captive animals and their earlier maturity have been attributed to nutritious diets in captivity, such as greater protein levels in early development, which have positive effects on skeletal growth [24].

Studies examining changes in morphology among wild and captive populations over multiple generations of animals have found significant differences. McPhee [7] found that captive populations of Oldfield mice (Peromyscus polionotus) had significantly smaller crania than wild populations, an effect compounded over generations. Another study [25] found that among feral American minks (Neogale vison), there were significant differences between the founder captive population and the following generations after 30–40 years in the wild (naturalized populations). It is unclear, however, to what extent the observed changes in morphology between captive and wild populations of the same species are owing to phenotypic plasticity within a lifetime or genetic selection.

Most studies comparing captive and wild animals have focused on the study of cranial dimensions and size, with only a few investigations of postcranial morphology [11,16]. One would expect to observe effects of captivity on the postcranial skeleton owing to confinement to small spaces, resulting in reduced mobility, unnatural substrates including a lack of trees for arboreal taxa, a lack of predatory behaviour and the absence of intraspecies interactions, all of which should reduce mechanical loading. Many experimental and observational studies have shown that bones respond to changes in mechanical loading during life, increasing in mass and strength with increased loading, and decreasing with reduced loading [26–37]. Consequently, captivity should result in a decrease in these parameters in comparison to wild counterparts. Trabecular bone in particular increases in density with increased biomechanical stress ([26–28,34,38–41]; also see Kivell [42] for a complete review), yet it has not been compared in captive and non-captive animals.

Few studies have examined postcrania of wild and captive animals. Canington et al. [11] compared cortical bone in captive and non-captive gorillas (Gorilla spp.) and found that the two groups differed in diaphyseal cross-sectional shape as a result of differences in locomotor behaviour resulting from environmental and associated locomotor differences. Harbers et al. [16] found that a biomechanically restricted environment altered calcaneal shape in wild boars (Sus scrofa), with reductions in mobility leading to altered use of the limb, which then resulted in shape change. Although the effects of captivity on cranial morphology in carnivores have been studied to a minor extent—e.g. lion crania [43,44], cheetah cranio-dental morphology and tiger cranial anatomy [44]—studies of carnivore postcrania are notably limited. Chirchir [45] compared trabecular bone density of domestic and wild canids and found that domestic dogs (Canis familiaris) exhibited significantly lower trabecular bone density (TBD) of the proximal femur and distal tibia than wild canids. Carnivores are good models for understanding differences in captive versus wild animals because they have differing home ranges and varying levels of cursoriality and arboreality in natural habitats, which are altered by captivity. Carnivores in the wild also use their forelimbs in non-locomotor activities such as hunting, subduing prey and swimming.

Here we examine how postcranial skeletal morphology—specifically trabecular bone volume fraction (BVF, or TBD)—differs between four species of captive felids and their wild conspecifics. BVF is strongly correlated with bone's stiffness (Young's modulus) and strength [46,47], which makes it critical for absorbing energy in joints during loading [48]. We hypothesize the following: (i) captive species will exhibit lower TBD than in their wild conspecifics given reduced mobility that results from the confined spaces in which captive species live. This hypothesis is based on the principle that reduced physical activity leads to reduced bone deposition; (ii) in captivity, there will be a more marked reduction in TBD in species with larger rather than smaller home ranges in the wild. This is based on preliminary results suggesting that mammals with longer daily travel distances display greater TBD resulting from greater physical activity [49]; and (iii) forelimbs will show more of an effect of captivity than hindlimbs. This hypothesis hinges in part on kinematic research showing that non-primate mammals have greater vertical ground reaction forces on the forelimb than on the hindlimb during terrestrial locomotion [50,51]. Thus, reduction in mobility should have a greater effect on forelimb loading and TBD. In addition, since the forelimbs are used in prey capture in the wild, this should also lead to reduced forelimb loading and bone density in captive carnivores.

This study assesses bone functional adaption in captive and wild animals to identify the effects of an altered environment. Our results are relevant to palaeontologists, mammalogists and conservationists interested in understanding the morphological and physiological differences observed between captive

and wild representatives of the same species. This is in part because of concerns in using skeletal remains of captive animals for comparative studies, despite the fact that such animals may not truly represent the natural condition, behaviourally or morphologically [1–5,10,11,44,52]. However, owing to the limited availability of complete skeletons in museum collections, researchers often rely on such collections— be they captive or wild specimens or specimens of unknown provenance—for comparative studies. Additionally, conservationists have a stake in this research because of their interest in understanding morphological and physiological differences that could affect the fitness of endangered and threatened species reintroduced into their natural habitats. Therefore, understanding these differences between wild and captive populations of a species is critical to these concerns.

# 2. Material and methods

## 2.1. Samples

In this study, we investigate the TBD (i.e. trabecular BVF) in the femoral and humeral heads of four felid species—mountain lions (*Puma concolor)*, cheetahs (*Acinonyx jubatus*), leopards (*Panthera pardus*) and jaguars (*Panthera onca*) (table 1). These species were selected owing to their diversity of locomotor behaviour and home ranges, as described below in the samples section, and their relatively close phylogenetic relationship [61]. All four belong to the family Felidae, with the lineage leading to the cheetah (*Acinonyx*) and mountain lion (*Puma*) originating at about 6.7 Ma [61–63], and the lineage giving rise to the genus *Panthera*, which includes leopards and jaguars, originating at about 10.8 Ma [61,63].

The humeral and femoral heads were selected as they are both involved in locomotion and allow examination of potential differences between fore and hindlimbs based on kinematic evidence suggesting greater forces in the forelimb than in the hindlimb during terrestrial locomotion [50]. These limb epiphyses are also relatively large, with sufficient trabeculae for analyses, and since they are located proximally should reflect overall loading of the fore and hindlimbs. We obtained 5–9 wild and captive specimens of each species from the National Museum of Natural History (NMNH) at the Smithsonian Institution in Washington, DC, and from the American Museum of Natural History (AMNH) in New York (table 1). Captive specimens are identified as those animals that lived in zoos from birth and were donated to museums upon their death, whereas the wild specimens are those animals that were shot in the wild and their skeletons and skins curated by the museums. All specimens are adult. Although there is some evidence that terrestrial mammals are susceptible to osteoarthritis [64], care was taken to select specimens that did not visually show any form of bone disease. Many specimens were of unknown sex (table 1). There have been efforts to sex the domestic cat skeleton using skeletal remains [65]; however, this has not been tested in wild felids. Therefore, data from both sexes were pooled in analyses. We did not test for differences between the sexes given the small number of known-sex individuals, which would greatly weaken statistical power.

### 2.1.1. Locomotor behaviour in the wild of study samples

Detailed locomotor behaviour is provided below; in summary, cheetahs are cursorial though with short daily travel distances and home range size; mountain lions are endurance travellers as inferred from their daily travel distance and large home ranges; leopards are adapted for arboreality and cursoriality while having short daily travel distance and home range size; jaguars are adept ambush hunters and swimmers with long daily travel distance and home range (table 1).

Mountain lions have great leaping power and are not adapted for running long distances [66]; they ambush their prey during hunting [67,68]. They do not run often, although they can attain speeds of up to 55 km h$^{-1}$. Their home range varies depending on the season; about 250 km$^2$ has been reported as a common average [53], although it can be greater. Mountain lions walk up to 6.6 km d$^{-1}$ [69]. Cheetahs are specialized for speed both anatomically and physiologically [70–72]. They can run at speeds of approximately 105–110 km h$^{-1}$ [72–75]. Cheetahs stalk their prey and charge from close proximity. Reported average daily movement for both males and females is 4 km d$^{-1}$ [69]. Some home range reports measure 50–130 km$^2$ [76], but this may be greater, e.g. 668 km$^2$ in Botswana depending on proximity to farmlands and competition from other large carnivores [77].

Leopards are strong and skilled climbers [66]. They walk slowly and stealthily but can also run briefly at up to 60 km h$^{-1}$. They are tree climbers especially when pursued by other carnivores [53]. The hunting style of leopards involves stalking and attacking prey from close quarters. Their reported home range size

**Table 1.** Samples of wild and captive specimens.

| species | home range and body mass | sex | museum collection | femoral head | humeral head |
|---|---|---|---|---|---|
| mountain lion (*Puma concolor*) | [c]250 km² [g]60 kg | 3 males, 6 females, 5 unknowns | [a]NMNH, Smithsonian | 8 wild, 6 captive; total = 14 pQCT scanned | 8 wild, 6 captive; total = 14 pQCT scanned |
| cheetah (*Acinonyx jubatus*) | [d]50–130 km² [h,i]52 kg | 5 males, 7 females, 3 unknowns | [a]NMNH, Smithsonian | 9 wild, 6 captive; total = 15 pQCT scanned | 9 wild, 6 captive; total = 15 pQCT scanned |
| leopard (*Panthera pardus*) | [e]8–63 km² [g,h]50 kg | 9 males, 3 females, 1 unknown | [a]NMNH, Smithsonian | 9 wild, 5 captive; total = 14 pQCT scanned | 9 wild, 5 captive; total = 14 pQCT scanned |
| jaguar (*Panthera onca*) | [f]25–38 km²/15–54 km² [h]85 kg | 3 males, 9 unknowns | [a]NMNH, Smithsonian and [b]AMNH, New York | 6 wild, 5 captive; total = 11 7 microCT scanned 4 pQCT scanned | 5 wild, 5 captive; total = 10 6 microCT scanned 4 pQCT scanned |

[a]NMNH captive specimens are from the National Zoo, Washington, DC.
[b]AMNH captive specimens are from the New York Zoological Society, NY.
[c]Hornocker [53].
[d]Kingdon [54].
[e]Mizutani & Jewell [55].
[f]Schaller & Crawshaw [56]; two ranges shown are for different regions (see text). Average body mass for pooled males and females is provided.
[g]Van Valkenburgh [57].
[h]Walker *et al.* [58].
[i]Macdonald [59].
[j]Eisenberg *et al.* [60].

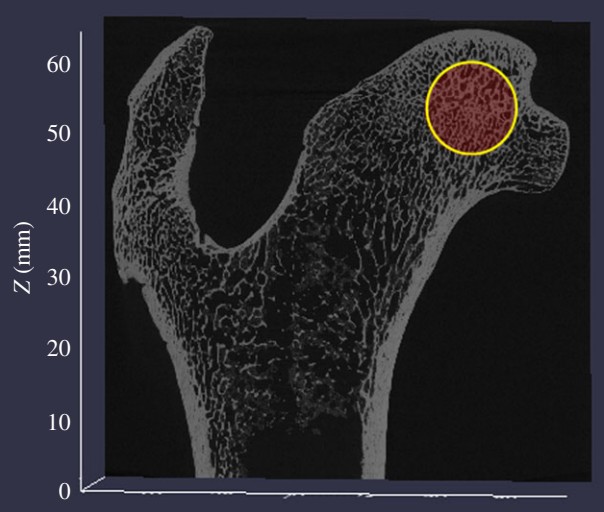

**Figure 1.** Example of VOI highlighted in red on the femoral head of a jaguar.

is 8–63 km$^2$, although this may vary depending on park size and the presence of domestic prey. For instance, there are reports of 14 km$^2$ for females and 3.28 km$^2$ for males in a 200 km$^2$ ranch [55]. They travel about 6.8 km d$^{-1}$ [69]. Jaguars are solitary hunters; their prey preferences are similar to those of leopards. However, jaguars differ from leopards in being about twice the mass with proportionally shorter and more robust limb bones [78,79]. Their daily travel distance is about 11 km d$^{-1}$ [69], and their reported home range size varies depending on locality. There are reports of 25–38 km$^2$ in Paraguay and 15–54 km$^2$ in the Pantanal [56]. Jaguar home range sizes vary depending on biome, e.g. Atlantic rainforest versus Amazon versus Pantanal.

### 2.1.2. Data acquisition

Data were acquired using two methods to secure sufficient data for this study: (i) high-resolution microCT imaging, carried out at the University of Texas at Austin's high-resolution X-ray CT (HRXCT) facility. The bone specimens were scanned at a 38 µm resolution, which offers high resolution for bone visualization and profiling; and (ii) peripheral quantitative computed tomography (pQCT), using a Stratec Research XCT scanner, with a resolution of 100 µm. Although both methods use X-rays, they have some differences. The pQCT scanner measures bone density in mg cm$^{-3}$, while the microCT scanner assembles three-dimensional images, which then are analysed to estimate BVF. Despite this difference in data acquisition, we have established a reliable method to combine datasets from both imaging techniques [49,80–82], further described below.

*MicroCT scanning*. We scanned 13 specimens using the microCT scanner (table 1). Three-dimensional Images were inspected in Amira 6.2 (Thermo Fisher Scientific) to ensure the image did not contain unwanted artefacts or inclusions. We then used the program QUANT 3D™ [83] to quantify trabecular bone volume. We selected one spherical volume of interest (VOI) in each epiphysis (figure 1). The size of the VOI was scaled to 65–75% of the superior–inferior breadth of the femoral and humeral heads based on epiphyseal size. For instance, we scaled the VOI to 75% for jaguar humeri and femora and 65% for leopards. We used the iterative thresholding method [83,84] to create a true binary image of black (air) and white (bone). QUANT 3D then quantifies the number of white pixels considered to be bone and divides that by the total number of pixels in a VOI, resulting in a ratio known as bone volume/total volume, i.e. trabecular BVF.

*pQCT scanning*. We scanned 94 specimens using pQCT imaging. A pQCT scanner uses a bone-equivalent phantom to convert linear attenuation coefficients obtained from X-rays into bone mineral density in mg cm$^{-3}$ [85–87]. The pQCT scanner derives bone density by measuring the material in a given (three-dimensional) voxel area, and although the resulting images were a single slice, density was derived from the width, length and slice thickness. Prior to scanning, the femoral and humeral head medio-lateral (ML) and anteroposterior (AP) breadths were measured using linear calipers. Scans were then taken at half the ML breadth of the femoral head and half the AP breadth of the humeral head (figure 2; also see [45,86]). After scanning each specimen, a region of interest (ROI) of about

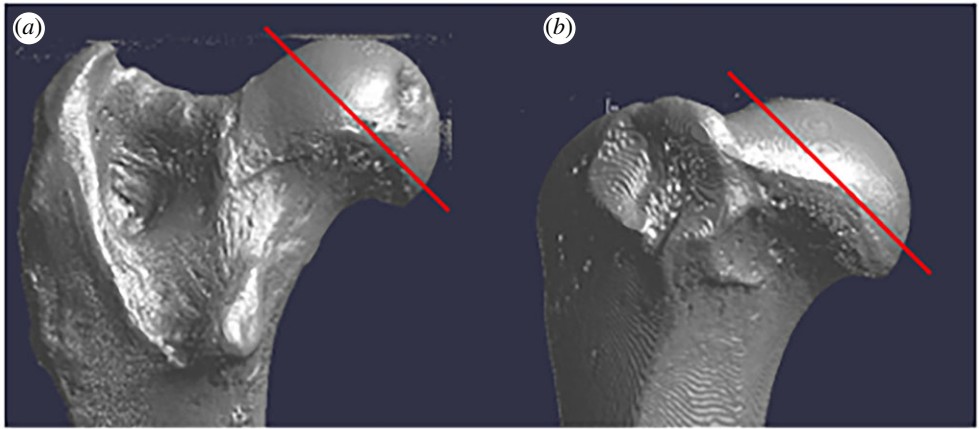

**Figure 2.** Red line indicates scanning location using the pQCT scanner in (*a*) femoral head and (*b*) humeral head.

65–75% (depending on epiphyseal size; the larger the epiphysis the greater the ROI) of the total slice cross-sectional area containing trabecular bone only was selected using the inbuilt applications Contour Mode and Peel Mode 1 in the pQCT scanner software. TBD values obtained from the pQCT scanner were then converted to BVF values, similar to those obtained from the microCT scanned images, using a linear regression equation obtained from previous work: $BVF = 0.006 \times TBD + 0.1567$ ($p < 0.001$; $r = 0.66$) [80].

### 2.1.3. Analyses

Unlike some trabecular bone properties, BVF does not scale allometrically with joint size in mammals [88]; therefore, there was no need to standardize it for body size. Our data were normally distributed as revealed by Shapiro–Wilk normality tests. We used boxplots and a scatter plot to visualize ranges of variation in BVF. We used two-way ANOVA for each skeletal element (femur, humerus) to identify the effects of origin (captivity versus wild) and species on BVF. To further investigate possible differences in effects on the fore and hindlimb, we carried out similar two-way ANOVAs of the natural log-transformed ratio of humerus/femur BVF (for justification for log-transformation of ratios; see Ruff [89]. One-way ANOVA and pairwise post hoc Tukey tests between species were also carried out for the humerus/femur ratio within wild and captive groups. A significance level of 0.05 was employed throughout. All statistical analyses and plotting were carried out using the statistical program R v. 3.6.1 [90].

## 3. Results

Mean values for BVF of wild and captive individuals in the forelimb and hindlimb within species are shown in table 2, and results from two-way ANOVA of each element are given in table 3. Boxplots for the femur and humerus are shown in figures 3 and 4. In both elements, captivity has a significant effect, with captive animals having lower BVF. The effect is stronger in the humerus ($p < 0.000001$) than in the femur ($p = 0.02$). Species has no significant effect (table 3). There is also no significant interaction between species and captivity/wild for either element (data not shown). However, it is apparent from the examination of distributions (figure 3) that leopards depart from other species in not exhibiting lower BVF values in the femur in captive animals.

A boxplot of the log humerus/femur BVF ratio is shown in figure 5, with results of the two-way ANOVA given in table 3. Captivity has a significant effect ($p < 0.005$), with captive animals exhibiting a lower ratio (figure 5). Species is also a significant factor ($p = 0.04$). However, when tested using one-way ANOVA within wild and within captive groups, interspecies differences are only significant within wild animals ($p = 0.03$), with captive animals showing no consistent trend ($p = 0.43$). Among wild animals, there is a downward trend from mountain lions to cheetahs to leopards to jaguars (figure 5), although Tukey tests are significant only between mountain lions and jaguars ($p = 0.02$). Figure 6 illustrates that average home range size (table 1) and humerus/femur BVF are positively correlated ($p = 0.008$) among wild animals; the correlation for captive animals is non-significant ($p = 0.17$).

**Table 2.** Mean BVF of captive and wild specimens in each element with standard error of the mean in parenthesis.

| species | femoral head | | humeral head | | |
| | captive (zoo) mean | wild-caught mean | country of origin for the wild-caught | captive (zoo) mean | wild-caught mean |
| --- | --- | --- | --- | --- | --- |
| mountain lion (P. concolor) | 0.426 (0.017) | 0.495 (0.009) | USA, Guatemala | 0.281 (0.01) | 0.374 (0.012) |
| cheetah (A. jubatus) | 0.459 (0.048) | 0.534 (0.018) | Kenya, Mozambique | 0.304 (0.02) | 0.369 (0.01) |
| leopard (P. pardus) | 0.493 (0.048) | 0.524 (0.026) | Tanzania, Mozambique, Kenya, Uganda, China, | 0.306 (0.017) | 0.344 (0.022) |
| jaguar (P. onca) | 0.451 (0.05) | 0.559 (0.02) | Brazil, Mexico | 0.261 (0.01) | 0.343 (0.01) |

**Table 3.** Two-way ANOVA of effects of origin (captive/wild) and species on BVF and the effects of origin (captive/wild) and species on log(humeral/femoral) BVF ratio.

| femur | $p$-value: effect of origin and species on BVF |
| --- | --- |
| origin | 0.0175 |
| species | 0.2196 |
| **humerus** | |
| origin | <0.000001 |
| species | 0.216 |
| **variable** | $p$-value: effect of origin and species on log humeral/humeral BVF ratio |
| origin | 0.00539 |
| species | 0.03616 |

There is no consistent relationship between home range size and BVF ratio between wild and captive animals, for either skeletal element (figure 7).

## 4. Discussion

In this study, we quantified BVF (i.e. TBD) in the femoral and humeral heads of captive and wild representatives of four felid species. We hypothesized that: (i) captive animals would exhibit lower BVF than their wild conspecifics given their reduced mobility owing to the confined spaces in which they live; (ii) there would be a more marked reduction in BVF in species with larger rather than smaller home ranges in the wild; and (iii) forelimbs would show more of an effect of captivity than hindlimbs. Our hypotheses are based on the understanding that decreased activity (mechanical loading) will lead to decreased bone deposition, as demonstrated by experimental and observational research [26–29,34,40]. Furthermore, because ground reaction forces are higher on the forelimbs during terrestrial locomotion in non-primate mammals [50,51], and because the forelimbs are used in prey capture, they should exhibit a greater effect owing to captivity than the hind limbs.

Our results support the first and third hypotheses. We found a significant effect of captivity on BVF, with both limbs exhibiting lower values in captive than in wild animals. As predicted, the effect is stronger in the humeral than in the femoral head. Species with larger home ranges in the wild also show a greater difference between humeral and femoral BVF (humerus greater), while there are no interspecies differences in humeral/femoral BVF among captive animals, providing some support for hypothesis (ii). However, there is no consistent relationship between home range size and the average difference in femoral or humeral BVF between captive and wild animals.

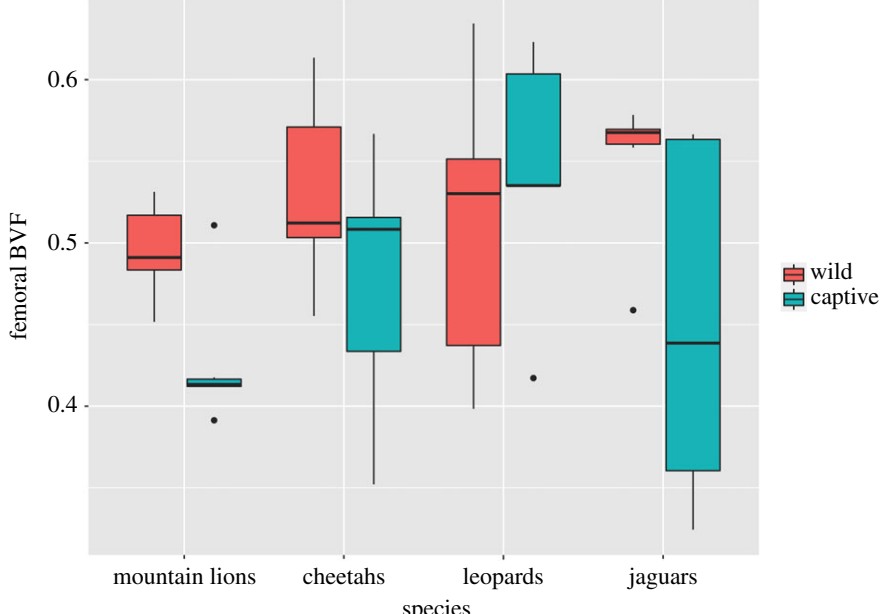

**Figure 3.** Boxplot showing BVF in the femoral head across species and between wild and captive specimens. The bold line represents the median, the box shows the interquartile range, and whiskers are 1.5 times the interquartile range. Black dots represent data points out of range from the rest. Significant differences are observed between wild and captive samples ($p < 0.05$) except among leopards. No significant differences were found across species.

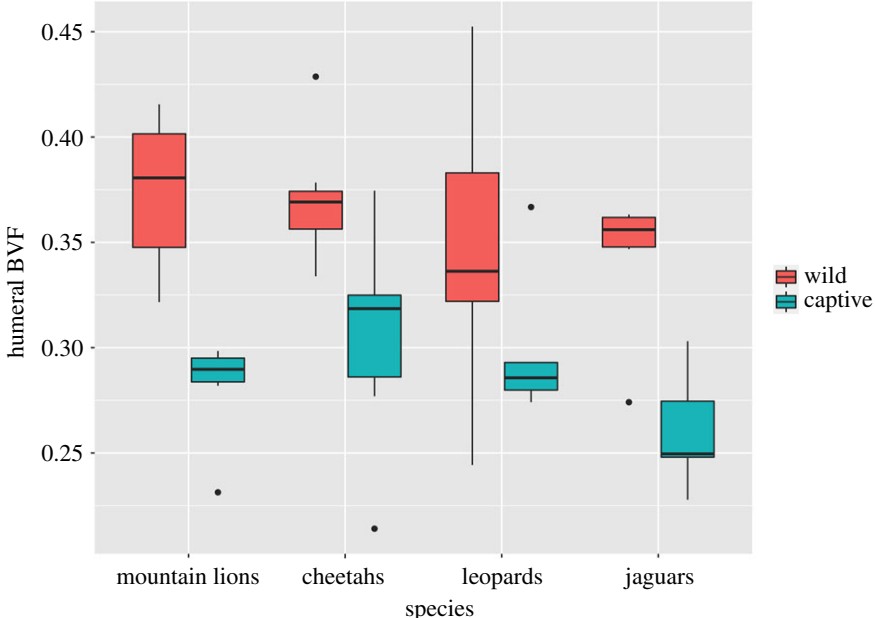

**Figure 4.** Boxplot showing BVF in the humeral head across species and between wild and captive specimens. The bold line represents the median, the box shows the interquartile range and whiskers are 1.5 times the interquartile range. Black dots represent data points out of range from the rest. Significant differences are observed between wild and captive samples ($p < 0.05$). No significant differences were found across species.

Previous studies have shown some significant differences in skeletal structure between captive and wild animals of the same species, mainly in cranial features. Alterations in mechanical loading of cranio-dental structures resulting from soft diets and improved caloric intake have been the primary explanations in these studies. Among pantherines, living in captivity had a significant effect on skull shape, such as a constricted foramen magnum and wide zygomatic arches attributed to soft diets [44]. There are also reports of greater cranial thickness and larger crania among captive animals compared to wild conspecifics [13,18,43,91] and an increase in body size in captivity ([6,8,10,20,92]; but also see [21,22,93,94]).

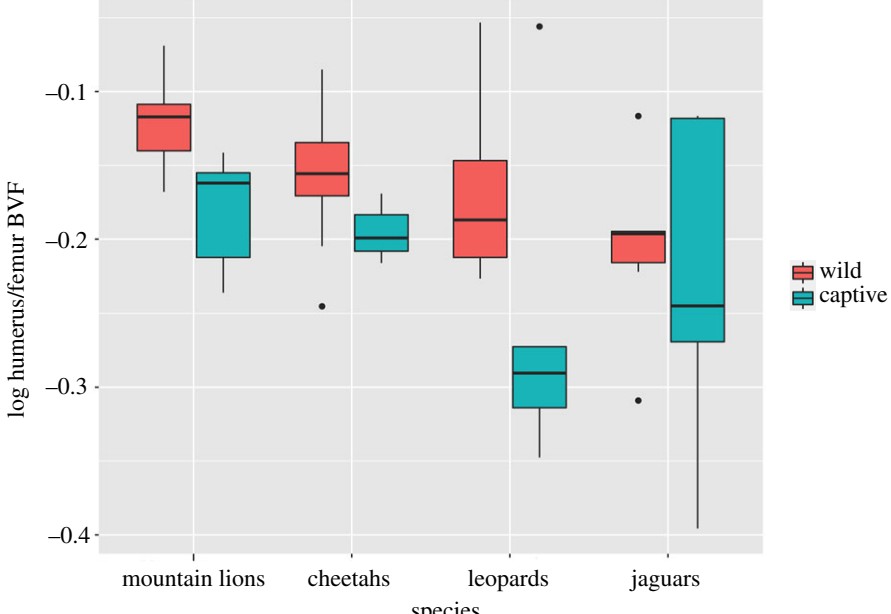

**Figure 5.** Boxplot showing log humeral/femoral BVF ratios of the wild and captive species. Shows a positive correlation between home range size and log humerus/femur BVF ratio. The bold line represents the median, the box shows the interquartile range and whiskers are 1.5 times the interquartile range. Black dots represent data points out of range from the rest.

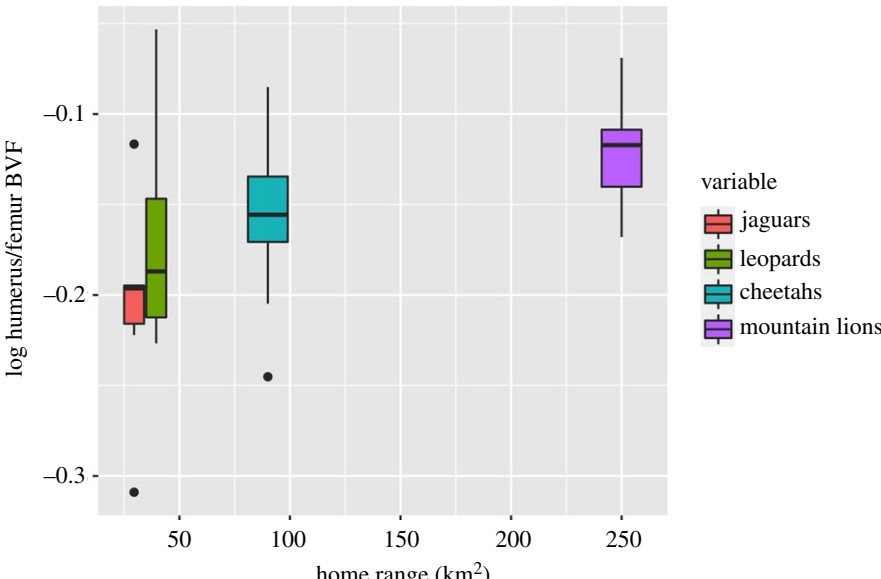

**Figure 6.** Boxplot of log humerus/femur BVF ratio of wild animals and home range size for each species. There is a positive correlation between the two ($p = 0.0008$), with smaller ratios among those with small home ranges and larger ratios among those with large home ranges. Black dots represent data points out of range from the rest.

Only a few studies have compared the postcranial skeletons of captive versus wild animals [11,16,93]. Some significant changes associated with captivity in calcaneal morphology in pigs [16] and in long bone cross-sectional shape in gorillas [11] have been documented. Domestication of dogs had a negative effect on BVF of the proximal femur and distal tibia when compared to wild canids [45]. The present study is, to our knowledge, the first to compare limb bone BVF in captive and wild conspecifics, and the first to examine felids. Our results are consistent with Chirchir [45] in showing a reduction in BVF in captives and also demonstrate some site-specific effects in the skeleton.

In addition to their involvement in terrestrial locomotion, the forelimbs of wild felids are used in diverse ways. Felids typically use their limbs for predation, and the claws are used to restrain prey [69,95]. Mountain lions are adept ambushing predators, which means they use their forelimbs for

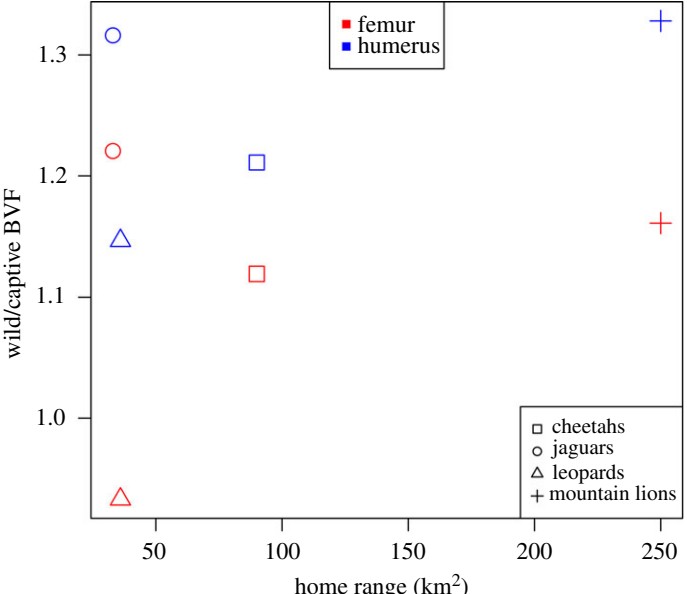

**Figure 7.** Scatter plot of BVF ratio between wild and captive animals and home range size for each element. There is no correlation for either bone.

grasping and subduing prey and thus potentially generate substantial loadings on them during predation. Leopards and jaguars are also ambush predators specialized for hunting in closed habitats and have sufficient strength to subdue prey on the ground [96]. Cheetahs use their forelimbs to subdue prey but because they lack the strength of other felids, they throw their prey off balance while running at high speed, use their retractile claws to restrain prey [95,97] and then use their jaws to grasp the trachea or snout to suffocate it [79]. Obviously, none of these behaviours would be practised by captive animals, which would preferentially affect the forelimb. Together with reduced overall mobility, which may have greater effects on forelimb loadings because of the greater ground reaction forces on the forelimb during terrestrial locomotion, these activities should lead to more marked reductions in BVF in the humerus than the femur among captives, as found in our study. The lack of a significant difference between captive and wild leopards in the femur is unexpected; however, it is plausible that owing to their fairly small home range (8–63 km$^2$), restricting the mobility of this animal in captivity is not sufficient to produce significant effects given that they are not travelling long distances even in the wild.

Our second hypothesis, which expects a greater reduction in BVF in species with larger home ranges, was generally not supported, except indirectly through the association of home range size with humeral/femoral BFV ratios. This finding may in part reflect how movement patterns in carnivores are characterized. For example, cheetahs have a daily travel distance of 4 km d$^{-1}$ [69] but a much larger home range size of 50–130 km$^2$ [76], whereas a jaguar travels 11 km d$^{-1}$ [69] and has a home range size of only up to 54 km$^2$ [56]. We relied on home range size rather than daily travel distance as an indicator of locomotor activity because it should reflect overall mobility over an extended time period. The finding that there is no correlation between the effects of captivity on BVF and home range size may not be surprising because most of these species have daily travel distances that are not highly diverse, ranging from 4 to 11 km d$^{-1}$. Additionally, as noted earlier, home range size in carnivores differs even within species depending on season and resource availability, including food, water and mates [98–100]. Finally, on a finer level, locomotor and non-locomotor behaviour varies in complex ways between taxa: mountain lions are adapted for endurance travel; cheetahs are cursorial; leopards are semi-arboreal, ambush predators and cursorial; and jaguars are arboreal, adapted for ambush hunting and swimming to catch prey. Although our statistical analyses indicated no species-specific effect on BVF, it is apparent that leopards do not show the expected captivity effect in the femur, in particular. More subtle behavioural differences between taxa (in the wild) may in part contribute to this finding, as may potential differences in captive environments.

Although the care of animals in zoos has improved considerably over the years, felids in captivity are still confined to fairly small spaces as compared to the wild. For instance, at the Columbus Zoo, a cheetah enclosure is only 429 m$^2$ (personal communications with zoo personnel), whereas in the wild, cheetahs

have home ranges of 50–130 km$^2$. There are indeed enrichment training programmes in some zoos to motivate animals to be mobile; these include the use of feeder mechanisms to encourage mobility, lure coursing and bungee cord jumping ([44]; and H. Chirchir 2021, personal observation). Despite these efforts, these environments cannot truly simulate living and hunting in the wild. The samples analysed in this study were acquired in the early to mid-twentieth century, possibly when zoos had not adopted current practices of enriching spaces to simulate natural habitats. Furthermore, the environment in which these animals lived is unknown, i.e. enclosure sizes and diets. Based on our results, captivity itself was the primary factor in determining variation in BVF between groups, with variation in home range size among wild taxa not contributing significantly to these patterns.

Dispersion of BVF values within species showed no consistent difference between captive and wild animals (figures 3 and 4). In the femur the coefficient of variation for BVF ranged from 5 to 15% in wild and 10–21% in captive animals, whereas in the humerus it ranged from 7 to 13% in wild and 10 to 22% in captive animals, thus showing great overlap. Several factors may have contributed to these variable results. A reduction in overall mobility among captives might be expected to produce less variation in BVF as activity levels are generally low among all captive animals. On the other hand, variation in artificial captive environments in terms of enclosure type, for example, could lead to more variability in mechanical loadings than among wild conspecifics. Variation in home range and environments among wild animals of the same species can also be considerable, e.g. see discussion of jaguars above.

In interpreting variation in BVF, other non-mechanical influences may also be considered, in particular nutritional differences. For instance, some zoos currently simulate natural diets by feeding felids with meat, while others rely on processed foods. This probably has different effects on skeletal anatomy, e.g. low calcium levels and pathologies as reported in some research on cranial morphology [18,21]. Other possible pathological, hormonal and physiological influences should also be considered [101,102]. Our study has some additional limitations. First, we did not analyse the architectural properties of trabecular bone (trabecular orientation, thickness and spacing), which have been shown to correlate with locomotor behaviour in mammals [41,103]. Our data consisted of both microCT- and pQCT-derived properties, and only microCT allows for the quantification of bone architecture. Using only the microCT data would have led to much smaller sample sizes, thus greatly diminishing statistical power. However, TBD as inferred from BVF alone is strongly predictive of bone strength and elasticity [47,104,105]. Second, sampling of additional skeletal locations would make our findings more robust. For example, including proximal and distal locations within limbs could yield interesting comparisons relevant to theories regarding trade-offs between limb element weight and strength [72,106,107]. Third, although the captive felids are known to have lived in zoos all their lives, we do not know the size or nature of the zoos' animal enclosures during their captivity. More information of this kind could help to clarify differences among the captive animals and possibly taxa as a whole. Finally, sex as a factor was not evaluated because sex was unidentified in many of the specimens, presenting a challenge in carrying out statistical comparisons.

Despite these caveats, our results add significantly to the existing literature by identifying differences between wild and captive animal populations in their postcranial morphology. They underscore the environmental and developmental plasticity of bone tissue. They also highlight the fact that parsing out morphological differences based on mobility among related wild species with overlapping home range size and daily travel distance is complex. This study informs research that involves *ex vivo* comparison of extant and extinct mammals, which depend on bone collections that could contain a mix of wild and captive animals. In addition, this study is of value to conservationists because morphological changes are important to consider when reintroducing captive endangered species to the wild. Morphological changes as a result of captivity may influence an individual's ability to survive in the wild, such as their ability to explore expansive home ranges, find mates and capture prey for food.

Data accessibility. Femoral and humeral BVF data are provided in the electronic supplementary material [108]. These data were obtained from three-dimensional images from microCT scanning and from single-slice images from pQCT scanning. Specimens were obtained from collections in the Department of Mammalogy, American Museum of Natural History and the Division of Mammals, National Museum of Natural History, Smithsonian Institution. All images are available for sharing from the principal investigator.
Authors' contributions. H.C.: conceptualization, data curation, formal analysis, funding acquisition, investigation, methodology, project administration, resources, software, supervision, validation, visualization, writing—original draft and writing—review and editing; C.R.: methodology, resources and writing—review and editing; K.M.H.: resources; R.P.: funding acquisition, resources, writing—review and editing.
All authors gave final approval for publication and agreed to be held accountable for the work performed therein.

Competing interests. We declare we have no competing interests.

Funding. The Peter Buck Postdoctoral Fellowship at NMNH and Marshall University research funds supported this research.

Acknowledgements. We thank Darrin Lunde for providing us access to specimens at the National Museum of Natural History, Smithsonian Institution, and Eleanor Hoeger and Marisa Surovy for providing us access to specimens at the American Museum of Natural History in New York. We thank Jessica Maisano and Mathew Colbert, for facilitating the microCT scanning at the HRXCT at the University of Texas at Austin. We also thank three reviewers for their thoughtful comments.

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
