## [Peer Review File · Royal Society Open Science]

Review History

RSOS-211345.R0 (Original submission)

Review form: Reviewer 1

Is the manuscript scientifically sound in its present form?

Yes

Are the interpretations and conclusions justified by the results?

Yes

Is the language acceptable?

Yes

Do you have any ethical concerns with this paper?

No

Have you any concerns about statistical analyses in this paper?

No

Recommendation?

Major revision is needed (please make suggestions in comments)

Comments to the Author(s)

This is very interesting work; I have some comments.

The introduction is too long; you should focus only on the relevant important information.

In method, how you make sure that all bone sample is a normal condition. Many diseases are affected by trabecular bone volume fraction such as Osteoarthritis. The previous study has been shown many samples in museums have OA, you should cite this article as well (Osteoarthritis in two marine mammals and 22 land mammals: learning from skeletal remains. *J Anat.* 2017 Jul;231(1):140-155. doi: 10.1111/joa.12620.). Please make inclusion and exclusion criteria more clear.

The other important issue when using samples from museums for this study. The process of skeleton performs. Most of the bone sample in the museum was boiled and sock in some chemical for cleaning such as Sodium hydroxide. These can be affected to bone structure, of cause trabecular bone volume fraction should be got some effect.

As known that sex is directly affected to bone morphology in feline (Determination of whether morphometric analysis of vertebrae in the domestic cat (*Felis catus*) is related to sex or skull shape. *Anat Sci Int.* 2020 Jun;95(3):387-398. doi: 10.1007/s12565-020-00533-3, Can feline (*Felis catus*) flat and long bone morphometry predict sex or skull shape? *Anat Sci Int.* 2019 Jun;94(3):245-256. doi: 10.1007/s12565-019-00480-8, Morphometric analysis of cervical vertebrae in some marine and land mammals. *Anat Histol Embryol.* 2021 Jul 17. doi: 10.1111/ahe.12725.). But, you had to pool the sample between male and female. Why do you use some technique to identify the sex either DNA or morphology? Please comment and discussion on this point, and do not forget cite those references.

The major issues that you classify into 2 groups for the study are the captive and wild animals. How do you classify this in particular "captive". How do you define "captive", it means this animal bone in the zoo? Some animals were caped from the wild and bring to the captive in the zoo, what do define? Wild or captive? Please make it clear in your manuscript.

Why only Jaguar was performing a microCT scanned?

Why do you perform only two parts, the femoral head, and humeral head? Why do you think this part is more important than the other?

Too much reference.

Review form: Reviewer 2 (Tracy Kivell)

Is the manuscript scientifically sound in its present form?

Yes

Are the interpretations and conclusions justified by the results?

Yes

Is the language acceptable?

Yes

Do you have any ethical concerns with this paper?

No

Have you any concerns about statistical analyses in this paper?

No

Recommendation?

Accept with minor revision (please list in comments)

Comments to the Author(s)

This paper elegantly tests the simple hypothesis that if bone responds to higher mechanical loads, then the bones of wild animals (felids) should show increased bone density/strength relative to their captive counterparts. The authors quantify/analyse trabecular bone volume fraction (TBVF) in the proximal femur and humerus in four felids that vary in their wild home ranges and locomotor/prey capture behaviours. This is a novel study. Variation in TBVF has not been studied before in captive vs. wild conspecifics and there are few studies of how morphology varies between captive/wild contexts in the postcranial skeleton in general. Introduction is clear and concise. Hypotheses are clearly articulated. The results are clearly presented and discussed. The results are interpreted with caution and the conclusions are well supported. There are impressively few typos. Overall, there is very little I can critique about this study or the manuscript; it was a joy to read and the results are exciting! The results are relevant to a broad range of disciplines, ranging from those interested in morphology/bone functional adaptation to animal conservation and thus appropriate for RS Open Science.

I list a few minor comments below. My only 'major' suggestion is this: I am struck by the differences in the degree of variation across some taxa between wild vs. captive values and between the femur and humerus. For example, the highly variable jaguar captive femur vs the wild femur, while the opposite is true in the mountain lions. It might be interesting to explore this variation in more detail (e.g. coefficient of variation) or to just discuss it in the Discussion. I would personally expect captive specimens might be more variable because enclosures, diet, activity levels, etc. would likely differ substantially across zoos/time period while, although wild animals might vary in frequencies of locomotion, terrain, etc., I would think their TBVF might be more constrained due to a more consistent/natural diet, activity levels, etc. However, this does not necessarily seem to be the case, which is quite interesting.

Minor comments:

Why use abbreviation 'TBVF' when the vast majority of studies use BV/TV and it seems the two are considered equivalent here (pg 4., line 24: "resulting in a ratio known as bone volume/total volume (BV/TV), i.e., trabecular bone volume fraction, TBVF." Is this because the pQCT bone mineral density values need to be converted to BV/TV values and the authors wish to make this distinction clear? If not, I would suggest using 'BV/TV' so it is clear that this study is analysing the same information as that other trabecular studies (including the Doube et al. 2011 study the authors cite in relation to the lack of significant allometric scaling).

pg 4/line 20: Why is the VOI scaled at 65-75% and not one standard scaling (e.g. all at 65%)? Does 65-75% mean it was 65% in some species and 75% in others, or 65-75% across all individuals depending on individual variation in the internal/external morphology?

Despite lack of allometric scaling in TBVF/BTV, it would be helpful to know the rough estimates of body mass in each of the four felid species (either wild, captive or both if known) just

so the reader has a better sense of the variation in body size across these four felids. Although I think most readers have a general sense of what these animals look like, it's harder to fully grasp their variation in body mass.

pg 6, line 26: might be worth clarifying here that this was only in the femur. The leopard humerus was significantly different.

Figure 1. Is it possible to highlight the VOI in a different colour (e.g. yellow) as the red is difficult to see against the black background

Table 1. remove 'all' from the mountain lion, femoral head column 'all pQCT scanned' (or add 'all' to the other columns/species for consistency)

Tracy Kivell

Decision letter (RSOS-211345.R0)

Dear Dr Chirchir

The Editors assigned to your paper RSOS-211345 "Effects of reduced mobility on trabecular bone density in captive felids" have now received comments from reviewers and would like you to revise the paper in accordance with the reviewer comments and any comments from the Editors. Please note this decision does not guarantee eventual acceptance.

Please submit your revised manuscript and required files (see below) no later than 21 days from today's (ie 20-Sep-2021) date. Note: the ScholarOne system will 'lock' if submission of the revision is attempted 21 or more days after the deadline. If you do not think you will be able to meet this deadline please contact the editorial office immediately.

on behalf of Dr Marco Palanca (Associate Editor) and Kevin Padian (Subject Editor)
 openscience@royalsociety.org

Subject Editor Comments to Author (Professor Kevin Padian):
 Comments to the Author:

Thank you for your submission. The reviewers are overall positive but they do have some comments that we ask you to address in your resubmission. Best wishes.

Reviewer comments to Author:

Reviewer: 1

Comments to the Author(s)

This is very interesting work; I have some comments.

The introduction is too long; you should focus only on the relevant important information.

In method, how you make sure that all bone sample is a normal condition. Many diseases are affected by trabecular bone volume fraction such as Osteoarthritis. The previous study has been shown many samples in museums have OA, you should cite this article as well (Osteoarthritis in two marine mammals and 22 land mammals: learning from skeletal remains. *J Anat.* 2017 Jul;231(1):140-155. doi: 10.1111/joa.12620.). Please make inclusion and exclusion criteria more clear.

The other important issue when using samples from museums for this study. The process of skeleton performs. Most of the bone sample in the museum was boiled and sock in some chemical for cleaning such as Sodium hydroxide. These can be affected to bone structure, of cause trabecular bone volume fraction should be got some effect.

As known that sex is directly affected to bone morphology in feline (Determination of whether morphometric analysis of vertebrae in the domestic cat (*Felis catus*) is related to sex or skull shape. *Anat Sci Int.* 2020 Jun;95(3):387-398. doi: 10.1007/s12565-020-00533-3, Can feline (*Felis catus*) flat and long bone morphometry predict sex or skull shape? *Anat Sci Int.* 2019 Jun;94(3):245-256. doi: 10.1007/s12565-019-00480-8, Morphometric analysis of cervical vertebrae in some marine and land mammals. *Anat Histol Embryol.* 2021 Jul 17. doi: 10.1111/ahe.12725.). But, you had to pool the sample between male and female. Why do you use some technique to identify the sex either DNA or morphology? Please comment and discussion on this point, and do not forget cite those references.

The major issues that you classify into 2 groups for the study are the captive and wild animals. How do you classify this in particular "captive". How do you define "captive", it means this animal bone in the zoo? Some animals were caped from the wild and bring to the captive in the zoo, what do define? Wild or captive? Please make it clear in your manuscript.

Why only Jaguar was performing a microCT scanned?

Why do you perform only two parts, the femoral head, and humeral head? Why do you think this part is more important than the other?

Too much reference.

Reviewer: 2

Comments to the Author(s)

This paper elegantly tests the simple hypothesis that if bone responds to higher mechanical loads, then the bones of wild animals (felids) should show increased bone density/strength relative to their captive counterparts. The authors quantify/analyse trabecular bone volume fraction (TBVF) in the proximal femur and humerus in four felids that vary in their wild home ranges and locomotor/prey capture behaviours. This is a novel study. Variation in TBVF has not been studied before in captive vs. wild conspecifics and there are few studies of how morphology varies between captive/wild contexts in the postcranial skeleton in general. Introduction is clear and concise. Hypotheses are clearly articulated. The results are clearly presented and discussed. The results are interpreted with caution and the conclusions are well supported. There are impressively few typos. Overall, there is very little I can critique about this study or the manuscript; it was a joy to read and the results are exciting! The results are relevant to a broad range of disciplines, ranging from those interested in morphology/bone functional adaptation to animal conservation and thus appropriate for RS Open Science.

I list a few minor comments below. My only 'major' suggestion is this: I am struck by the differences in the degree of variation across some taxa between wild vs. captive values and between the femur and humerus. For example, the highly variable jaguar captive femur vs the wild femur, while the opposite is true in the mountain lions. It might be interesting to explore this variation in more detail (e.g. coefficient of variation) or to just discuss it in the Discussion. I would personally expect captive specimens might be more variable because enclosures, diet, activity levels, etc. would likely differ substantially across zoos/time period while, although wild animals might vary in frequencies of locomotion, terrain, etc., I would think their TBVF might be more constrained due to a more consistent/natural diet, activity levels, etc. However, this does not necessarily seem to be the case, which is quite interesting.

Minor comments:

Why use abbreviation 'TBVF' when the vast majority of studies use BV/TV and it seems the two are considered equivalent here (pg 4., line 24: "resulting in a ratio known as bone volume/total volume (BV/TV), i.e., trabecular bone volume fraction, TBVF." Is this because the pQCT bone mineral density values need to be converted to BV/TV values and the authors wish to make this distinction clear? If not, I would suggest using 'BV/TV' so it is clear that this study is analysing the same information as that other trabecular studies (including the Doube et al. 2011 study the authors cite in relation to the lack of significant allometric scaling).

pg 4/line 20: Why is the VOI scaled at 65-75% and not one standard scaling (e.g. all at 65%)? Does 65-75% mean it was 65% in some species and 75% in others, or 65-75% across all individuals depending on individual variation in the internal/external morphology?

Despite lack of allometric scaling in TBVF/BVTV, it would be helpful to know the rough estimates of body mass in each of the four felid species (either wild, captive or both if known) just so the reader has a better sense of the variation in body size across these four felids. Although I think most readers have a general sense of what these animals look like, it's harder to fully grasp their variation in body mass.

pg 6, line 26: might be worth clarifying here that this was only in the femur. The leopard humerus was significantly different.

Figure 1. Is it possible to highlight the VOI in a different colour (e.g. yellow) as the red is difficult to see against the black background

Table 1. remove 'all' from the mountain lion, femoral head column 'all pQCT scanned' (or add 'all' to the other columns/species for consistency)

Tracy Kivell

===PREPARING YOUR MANUSCRIPT===

===PREPARING YOUR REVISION IN SCHOLARONE===

Author's Response to Decision Letter for (RSOS-211345.R0)

See Appendices A & B.

RSOS-211345.R1 (Revision)

Review form: Reviewer 2 (Tracy Kivell)

Is the manuscript scientifically sound in its present form?

Yes

Are the interpretations and conclusions justified by the results?

Yes

Is the language acceptable?

Yes

Do you have any ethical concerns with this paper?

No

Have you any concerns about statistical analyses in this paper?

No

Recommendation?

Accept as is

Comments to the Author(s)

The authors have done an excellent job of addressing my comments, which were relatively minor, and those of the other reviewer. This was a strong paper to begin with and the changes that have been implemented in the revised version have made it even stronger. I am happy for this manuscript to be accepted and look forward to seeing it published.

Tracy Kivell

Review form: Reviewer 3 (Luca Cristofolini)

Is the manuscript scientifically sound in its present form?

Yes

Are the interpretations and conclusions justified by the results?

Yes

Is the language acceptable?

Yes

Do you have any ethical concerns with this paper?

No

Have you any concerns about statistical analyses in this paper?

No

Recommendation?

Accept with minor revision (please list in comments)

Comments to the Author(s)

GENERAL COMMENTS

This paper investigates the effects of loading activities on bone remodeling by comparing felids in captivity and wild. The research question is clear and highly relevant. The study is nicely designed to address the research question. The methods are robust and the data are analyzed convincingly. I only have minor comments on this paper.

SPECIFIC COMMENTS

The paper is generally well-written and the images are clear. Here are some detailed comments:

1. INTRODUCTION, page 1, line 33: a closed bracket is missing
2. INTRODUCTION, page 2, line 31 and following: I wonder whether hypothesis (iii) was really formulated before initiating the study, or was it more general, e.g. you originally suspected "only" that differences might exist between bones
3. MATERIALS AND METHODS, PAGE 3: the information under the sub-heading "Locomotor behaviour in the wild" looks more like Introduction (or relevant to the discussion) than M&M
4. TABLE 3: the P-values are definitely relevant. Not so sure about the F-values

Decision letter (RSOS-211345.R1)

Dear Dr Chirchir

On behalf of the Editors, we are pleased to inform you that your Manuscript RSOS-211345.R1 "Effects of reduced mobility on trabecular bone density in captive felids" has been accepted for publication in Royal Society Open Science subject to minor revision in accordance with the referees' reports. Please find the referees' comments along with any feedback from the Editors below my signature.

Note that, unless there are strong reasons for not doing so (and these must be agreed with the editorial office), it is a requirement of acceptance that your datasets are accessible, per the Editor's request.

Please submit your revised manuscript and required files (see below) no later than 7 days from today's (ie 16-Feb-2022) date. Note: the ScholarOne system will 'lock' if submission of the revision is attempted 7 or more days after the deadline. If you do not think you will be able to meet this deadline please contact the editorial office immediately.

on behalf of Dr Marco Palanca (Associate Editor) and Kevin Padian (Subject Editor)
openscience@royalsociety.org

Associate Editor Comments to Author (Dr Marco Palanca):

Associate Editor: 1

Comments to the Author:

Please, add a statement about data availability for the scientific community: if data sharing is possible or if data are copyright of the museums.

Reviewer comments to Author:

Reviewer: 2

Comments to the Author(s)

The authors have done an excellent job of addressing my comments, which were relatively minor, and those of the other reviewer. This was a strong paper to begin with and the changes that have been implemented in the revised version have made it even stronger. I am happy for this manuscript to be accepted and look forward to seeing it published.

Tracy Kivell

Reviewer: 3

Comments to the Author(s)

GENERAL COMMENTS

This paper investigates the effects of loading activities on bone remodeling by comparing felids in captivity and wild. The research question is clear and highly relevant. The study is nicely designed to address the research question. The methods are robust and the data are analyzed convincingly. I only have minor comments on this paper.

SPECIFIC COMMENTS

The paper is generally well-written and the images are clear. Here are some detailed comments:

1. INTRODUCTION, page 1, line 33: a closed bracket is missing

2. INTRODUCTION, page 2, line 31 and following: I wonder whether hypothesis (iii) was really formulated before initiating the study, or was it more general, e.g. you originally suspected “only” that differences might exist between bones

3. MATERIALS AND METHODS, PAGE 3: the information under the sub-heading “Locomotor behaviour in the wild” looks more like Introduction (or relevant to the discussion) than M&M

4. TABLE 3: the P-values are definitely relevant. Not so sure about the F-values

===PREPARING YOUR MANUSCRIPT===

one version should clearly identify all the changes that have been made (for instance, in coloured highlight, in bold text, or tracked changes);

===PREPARING YOUR REVISION IN SCHOLARONE===

-- If you are requesting an article processing charge waiver, you must select the relevant waiver option (if requesting a discretionary waiver, the form should have been uploaded, see 'File upload' above).

-- If you have uploaded any electronic supplementary (ESM) files, please ensure you follow the guidance at <https://royalsociety.org/journals/authors/author-guidelines/#supplementary-material> to include a suitable title and informative caption. An example of appropriate titling and captioning may be found at https://figshare.com/articles/Table_S2_from_Is_there_a_trade-off_between_peak_performance_and_performance_breadth_across_temperatures_for_aerobic_scope_in_teleost_fishes_/3843624.

Author's Response to Decision Letter for (RSOS-211345.R1)

See Appendix C.

Decision letter (RSOS-211345.R2)

Dear Dr Chirchir,

It is a pleasure to accept your manuscript entitled "Effects of reduced mobility on trabecular bone density in captive felids" in its current form for publication in Royal Society Open Science.

on behalf of Dr Marco Palanca (Associate Editor) and Kevin Padian (Subject Editor)
openscience@royalsociety.org

Appendix A

Response to Reviewers

Reviewer comments to Author:

Reviewer: 1

Comments to the Author(s)

This is very interesting work; I have some comments.

The introduction is too long; you should focus only on the relevant important information.

We feel that the Introduction is important because it makes a strong case for why this study is significant and timely. It includes a thorough review of previous work on captive vs. wild comparisons of skeletal morphology, emphasizing the fact that no studies have investigated bone density in felid postcrania. We heeded the Reviewer's comment and made some changes to eliminate unnecessary phrasing, while most of the content was retained.

In method, how you make sure that all bone sample is a normal condition. Many diseases are affected by trabecular bone volume fraction such as Osteoarthritis. The previous study has been shown many samples in museums have OA, you should cite this article as well (Osteoarthritis in two marine mammals and 22 land mammals: learning from skeletal remains. *J Anat.* 2017 Jul;231(1):140-155. doi: 10.1111/joa.12620.). Please make inclusion and exclusion criteria more clear.

We have cited this reference with regard to the possibility that terrestrial mammals may have osteoarthritis. We took care in our samples to include only specimens that did not show any bone disease on visual inspection.

The other important issue when using samples from museums for this study. The process of skeleton performs. Most of the bone sample in the museum was boiled and sock in some chemical for cleaning such as Sodium hydroxide. These can be affected to bone structure, of cause trabecular bone volume fraction should be got some effect.

The reviewer is correct in noting that the use of sodium hydroxide to prepare skeletons may have an effect on bone density. The literature shows that it degrades tissue -mostly soft tissue, hence its effectiveness in cleaning bones (e.g., Onwuama et al. 2012; Dowell, 2012). However, after speaking with museum collection managers, we are informed that given the time period in which these bones were acquired - early to mid-20th century - they were certainly boiled and possibly varying concentrations of ammonia were used. It is unlikely that chemicals such as sodium hydroxide or sodium peroxide were used at the time. We sought the advice of the relevant collections managers because there are no records documenting the treatment method. Given the unlikely use of sodium hydroxide/peroxide, we do not anticipate any effects on bone tissue. During specimen selection, one could observe that the bones were fairly greasy suggesting that they were boiled.

As known that sex is directly affected to bone morphology in feline (Determination of whether morphometric analysis of vertebrae in the domestic cat (*Felis catus*) is related to sex or skull shape. *Anat Sci Int.* 2020 Jun;95(3):387-398. doi: 10.1007/s12565-020-00533-3, Can feline (*Felis catus*) flat and long bone morphometry predict sex or skull shape? *Anat Sci Int.* 2019 Jun;94(3):245-256. doi: 10.1007/s12565-019-00480-8, Morphometric analysis of cervical vertebrae in some marine and land mammals. *Anat Histol Embryol.* 2021 Jul 17. doi: 10.1111/ah.12725.). But, you had to pool the sample between male and female. Why do you use some technique to identify the sex either DNA or morphology? Please comment and discussion on this point, and do not forget cite those references.

We thank the Reviewer for these references. The two papers that investigate the association between sex and morphological traits are Boonsri et al. (2019 and (2020), both of which investigated long bone and vertebral traits to identify a potential association with sex in the domestic cat. These studies may or may not have a bearing on identifying sex in wild cats; because these studies were on *Felis catus*, the methods may not be directly applicable to our samples. Furthermore, Boonsri et al. (2020) did not find a relationship among vertebral measurements, skull shape and sex, thus using these traits would not help identify sex.

Additionally, some of our samples had known sexes from the time they were alive and a quick comparison of means between males and females of those of known sex did not show great divergence. For example, among cheetahs, the mean BVF between females and males in the humerus was 0.36 and 0.34, respectively, and in the femur, it was 0.52 and 0.51, respectively. This suggests a lack of significant sex differences, yet the question opens up a further line of inquiry for future study. We have cited Boonsri et al. (2019) indicating efforts to sex the skeleton from postcranial remains in domestic cats, but also that we do not know if they can be applied to wild cats.

At the end of our Discussion section, we also describe the fact that we did not evaluate sex as a study limitation: "Finally, sex as a factor was not evaluated because sex was unidentified in many of the specimens, presenting a challenge in carrying out statistical comparisons".

The major issues that you classify into 2 groups for the study are the captive and wild animals. How do you classify this in particular "captive". How do you define "captive", it means this animal bone in the zoo? Some animals were caged from the wild and brought to the captive in the zoo, what do you define? Wild or captive? Please make it clear in your manuscript.

Thank you for this comment. We describe captive vs. wild as follows in the second paragraph of Materials and Methods: "Captive specimens are identified as those animals that lived in zoos from birth and were donated to museums upon their death, whereas the wild specimens are those animals that were shot in the wild and their skeletons and skins curated by the museums". However, as the Reviewer correctly notes there are challenges in identifying captive animals that may have been captured in the wild and then kept in zoos. In corresponding with museum collections managers, as far as they are aware, the zoo animals were born in zoos, but given the lack of records documenting these for all of the specimens there is a small possibility that some of those captive animals could indeed initially have been captured in the wild initially, which we cannot rule out. For a future study including other specimens from other museums, we may be able to find better records of specimen acquisition that would help to better address this question.

Why only Jaguar was performing a microCT scanned? The cost of microCT imaging is still prohibitive to many researchers and microCT imaging is not readily accessible in many museums. Therefore, to collect sufficient data for this study we had to rely on both techniques. We have added a sentence to this effect in the "Data acquisition" section of the Materials & Methods.

Why do you perform only two parts, the femoral head, and humeral head? Why do you think this part is more important than the other?

We selected the two elements because they are both involved in locomotion and also to study differences between fore and hindlimbs. These epiphyses are fairly large, facilitating sampling of sufficient trabeculae. In the second paragraph of the Materials and Methods section, we offered

this explanation: “The humeral and femoral heads were selected as they are both involved in locomotion and allow examination of potential differences between fore and hindlimbs based on kinematic evidence suggesting greater forces in the forelimb than in the hind limb during terrestrial locomotion [50]. These limb epiphyses are also relatively large, with sufficient trabeculae for analyses, and since they are located proximally should reflect overall loading of the fore and hind limbs.”

Too much reference.

Each of the statements and arguments made especially in the Introduction and Discussion sections are based on previous work, which we find imperative to cite; hence the many references. We are reticent to pare down the references given that they form the foundation of our study.

Reviewer: 2

Comments to the Author(s)

This paper elegantly tests the simple hypothesis that if bone responds to higher mechanical loads, then the bones of wild animals (felids) should show increased bone density/strength relative to their captive counterparts. The authors quantify/analyse trabecular bone volume fraction (TBVF) in the proximal femur and humerus in four felids that vary in their wild home ranges and locomotor/prey capture behaviours. This is a novel study. Variation in TBVF has not been studied before in captive vs. wild conspecifics and there are few studies of how morphology varies between captive/wild contexts in the postcranial skeleton in general. Introduction is clear and concise. Hypotheses are clearly articulated. The results are clearly presented and discussed. The results are interpreted with caution and the conclusions are well supported. There are impressively few typos. Overall, there is very little I can critique about this study or the manuscript; it was a joy to read and the results are exciting! The results are relevant to a broad range of disciplines, ranging from those interested in morphology/bone functional adaptation to animal conservation and thus appropriate for RS **Open Science**.

I list a few minor comments below. My only ‘major’ suggestion is this: I am struck by the differences in the degree of variation across some taxa between wild vs. captive values and between the femur and humerus. For example, the highly variable jaguar captive femur vs the wild femur, while the opposite is true in the mountain lions. It might be interesting to explore this variation in more detail (e.g. coefficient of variation) or to just discuss it in the Discussion. I would personally expect captive specimens might be more variable because enclosures, diet, activity levels, etc. would likely differ substantially across zoos/time period while, although wild animals might vary in frequencies of locomotion, terrain, etc., I would think their TBVF might be more constrained due to a more consistent/natural diet, activity levels, etc. However, this does not necessarily seem to be the case, which is quite interesting.

We thank the Reviewer for raising this point. We have now included a new paragraph in the Discussion that specifically addresses this issue, including presentation of coefficients of variation of BVF in captive and wild samples. As we note, it is possible that there would be greater variation in captive than wild animals, as suggested by the Reviewer, due to variation in captive living conditions; it is also possible that less variation in the captive species would result from an overall reduction in mobility. Our comparison of CVs shows no consistent patterning and great overlap between wild and captive animals, possibly as a result of these conflicting influences.

Minor comments:

Why use abbreviation 'TBVF' when the vast majority of studies use BV/TV and it seems the two are considered equivalent here (pg 4., line 24: "resulting in a ratio known as bone volume/total volume (BV/TV), i.e., trabecular bone volume fraction, TBVF." Is this because the pQCT bone mineral density values need to be converted to BV/TV values and the authors wish to make this distinction clear? If not, I would suggest using 'BV/TV' so it is clear that this study is analysing the same information as that other trabecular studies (including the Doube et al. 2011 study the authors cite in relation to the lack of significant allometric scaling).

Thank you for the comment. This is a point that we have grappled with previously in comparing results from 3D images derived from microCT and those from a pQCT. Although we have successfully (Chirchir et al. 2015) found a correlation between the two measures, we thought it would still be important to use a term that shows that we are not strictly referring to BV/TV only, but also data from a pQCT scanner that have previously been identified as trabecular bone volume fraction (or just trabecular fraction). If we were to use BV/TV, it is still referring to bone volume fraction. Therefore, for consistency, we use BVF to refer to bone volume fraction derived from both microCT and pQCT data.

pg 4/line 20: Why is the VOI scaled at 65-75% and not one standard scaling (e.g. all at 65%)? Does 65-75% mean it was 65% in some species and 75% in others, or 65-75% across all individuals depending on individual variation in the internal/external morphology?.

We tried to select similar VOI sizes for both pQCT- and microCT-derived data. We scaled within a range 65-75% because of the variation in size of the different species and elements. So, for each species with similar epiphyseal size we selected the same size VOI; e.g., for all leopards in the humerus we used 65% VOI. For all jaguars we used a VOI of 75% in the femur. We have clarified this point in the text.

Despite lack of allometric scaling in TBVF/BVTV, it would be helpful to know the rough estimates of body mass in each of the four felid species (either wild, captive or both if known) just so the reader has a better sense of the variation in body size across these four felids. Although I think most readers have a general sense of what these animals look like, it's harder to fully grasp their variation in body mass.

We have provided male and female average body mass for each species in Table 1.

pg 6, line 26: might be worth clarifying here that this was only in the femur. The leopard humerus was significantly different.

Corrected

Figure 1. Is it possible to highlight the VOI in a different colour (e.g. yellow) as the red is difficult to see against the black background

Corrected

Table 1. remove 'all' from the mountain lion, femoral head column 'all pQCT scanned' (or add 'all' to the other columns/species for consistency)

Corrected

Tracy Kivell

Appendix B

November 10th 2021

Dear Editor,

Re: Manuscript revision and resubmission

Thank you for the opportunity to revise and resubmit manuscript ID number **RSOS-211345** titled "Effects of reduced mobility on trabecular bone density in captive felids". As you will see below, we have responded to reviewer comments point-by-point. Additionally, the revisions in the main text and in the tables are highlighted in blue for the readers' convenience.

We have revised the manuscript to reflect reviewers' recommendations and offered explanations in the instances that we did not effect a recommendation. These include performing additional basic statistical analysis recommended by Reviewer 2 and issues brought up by Reviewer 1 on the definition of "wild" vs "captive", sexing the felid skeletons and the preparation techniques used by museums. Reviewer 1 also thought the references were too many but we retained them as we thought it was necessary to exhaustively cover previous research

In the Introduction section, Reviewer 1 recommended shortening it while Reviewer 2 was complimentary of it. While we made some changes to the Introduction, for the most part we retained it as it was since we deemed it necessary to build/lay the background for the significance of our study.

The reviewers' comments have been very helpful in improving the manuscript and we are very grateful to them.

Again, thank you for considering our manuscript for publication in Royal Open Science. We look forward to hearing from you.

Sincerely,

Habiba Chirchir

Reviewer comments to Author:

Reviewer: 1

Comments to the Author(s)

This is very interesting work; I have some comments.

The introduction is too long; you should focus only on the relevant important information.

We feel that the Introduction is important because it makes a strong case for why this study is significant and timely. It includes a thorough review of previous work on captive vs. wild comparisons of skeletal morphology, emphasizing the fact that no studies have investigated bone density in felid postcrania. We heeded the Reviewer's comment and made some changes to eliminate unnecessary phrasing, while most of the content was retained.

In method, how you make sure that all bone sample is a normal condition. Many diseases are affected by trabecular bone volume fraction such as Osteoarthritis. The previous study has been shown many samples in museums have OA, you should cite this article as well (Osteoarthritis in two marine mammals and 22 land mammals: learning from skeletal remains. *J Anat.* 2017 Jul;231(1):140-155. doi: 10.1111/joa.12620.). Please make inclusion and exclusion criteria more clear.

We have cited this reference with regard to the possibility that terrestrial mammals may have osteoarthritis. We took care in our samples to include only specimens that did not show any bone disease on visual inspection.

The other important issue when using samples from museums for this study. The process of skeleton performs. Most of the bone sample in the museum was boiled and sock in some chemical for cleaning such as Sodium hydroxide. These can be affected to bone structure, of cause trabecular bone volume fraction should be got some effect.

The reviewer is correct in noting that the use of sodium hydroxide to prepare skeletons may have an effect on bone density. The literature shows that it degrades tissue -mostly soft tissue, hence its effectiveness in cleaning bones (e.g., Onwuama et al. 2012; Dowell, 2012). However, after speaking with museum collection managers, we are informed that given the time period in which these bones were acquired - early to mid-20th century - they were certainly boiled and possibly varying concentrations of ammonia were used. It is unlikely that chemicals such as sodium hydroxide or sodium peroxide were used at the time. We sought the advice of the relevant collections managers because there are no records documenting the treatment method. Given the unlikely use of sodium hydroxide/peroxide, we do not anticipate any effects on bone tissue. During specimen selection, one could observe that the bones were fairly greasy suggesting that they were boiled.

As known that sex is directly affected to bone morphology in feline (Determination of whether morphometric analysis of vertebrae in the domestic cat (*Felis catus*) is related to sex or skull shape. *Anat Sci Int.* 2020 Jun;95(3):387-398. doi: 10.1007/s12565-020-00533-3, Can feline (*Felis catus*) flat and long bone morphometry predict sex or skull shape? *Anat Sci Int.* 2019 Jun;94(3):245-256. doi: 10.1007/s12565-019-00480-8, Morphometric analysis of cervical vertebrae in some marine and land mammals. *Anat Histol Embryol.* 2021 Jul 17. doi: 10.1111/ah.12725.). But, you had to pool the sample between male and female. Why do you use some technique to identify the sex either DNA or morphology? Please comment and discussion on this point, and do not forget cite those references.

We thank the Reviewer for these references. The two papers that investigate the association between sex and morphological traits are Boonsri et al. (2019 and (2020), both of which

investigated long bone and vertebral traits to identify a potential association with sex in the domestic cat. These studies may or may not have a bearing on identifying sex in wild cats; because these studies were on *Felis catus*, the methods may not be directly applicable to our samples. Furthermore, Boonsri et al. (2020) did not find a relationship among vertebral measurements, skull shape and sex, thus using these traits would not help identify sex.

Additionally, some of our samples had known sexes from the time they were alive and a quick comparison of means between males and females of those of known sex did not show great divergence. For example, among cheetahs, the mean BVF between females and males in the humerus was 0.36 and 0.34, respectively, and in the femur, it was 0.52 and 0.51, respectively. This suggests a lack of significant sex differences, yet the question opens up a further line of inquiry for future study. We have cited Boonsri et al. (2019) indicating efforts to sex the skeleton from postcranial remains in domestic cats, but also that we do not know if they can be applied to wild cats.

At the end of our Discussion section, we also describe the fact that we did not evaluate sex as a study limitation: "Finally, sex as a factor was not evaluated because sex was unidentified in many of the specimens, presenting a challenge in carrying out statistical comparisons".

The major issues that you classify into 2 groups for the study are the captive and wild animals. How do you classify this in particular "captive". How do you define "captive", it means this animal bone in the zoo? Some animals were caped from the wild and bring to the captive in the zoo, what do define? Wild or captive? Please make it clear in your manuscript.

Thank you for this comment. We describe captive vs. wild as follows in the second paragraph of Materials and Methods: "Captive specimens are identified as those animals that lived in zoos from birth and were donated to museums upon their death, whereas the wild specimens are those animals that were shot in the wild and their skeletons and skins curated by the museums". However, as the Reviewer correctly notes there are challenges in identifying captive animals that may have been captured in the wild and then kept in zoos. In corresponding with museum collections managers, as far as they are aware, the zoo animals were born in zoos, but given the lack of records documenting these for all of the specimens there is a small possibility that some of those captive animals could indeed initially have been captured in the wild initially, which we cannot rule out. For a future study including other specimens from other museums, we may be able to find better records of specimen acquisition that would help to better address this question.

Why only Jaguar was performing a microCT scanned? The cost of microCT imaging is still prohibitive to many researchers and microCT imaging is not readily accessible in many museums. Therefore, to collect sufficient data for this study we had to rely on both techniques. We have added a sentence to this effect in the "Data acquisition" section of the Materials & Methods.

Why do you perform only two parts, the femoral head, and humeral head? Why do you think this part is more important than the other?

We selected the two elements because they are both involved in locomotion and also to study differences between fore and hindlimbs. These epiphyses are fairly large, facilitating sampling of sufficient trabeculae. In the second paragraph of the Materials and Methods section, we offered this explanation: "The humeral and femoral heads were selected as they are both involved in locomotion and allow examination of potential differences between fore and hindlimbs based on

kinematic evidence suggesting greater forces in the forelimb than in the hind limb during terrestrial locomotion [50]. These limb epiphyses are also relatively large, with sufficient trabeculae for analyses, and since they are located proximally should reflect overall loading of the fore and hind limbs.”

Too much reference.

Each of the statements and arguments made especially in the Introduction and Discussion sections are based on previous work, which we find imperative to cite; hence the many references. We are reticent to pare down the references given that they form the foundation of our study.

Reviewer: 2

Comments to the Author(s)

This paper elegantly tests the simple hypothesis that if bone responds to higher mechanical loads, then the bones of wild animals (felids) should show increased bone density/strength relative to their captive counterparts. The authors quantify/analyse trabecular bone volume fraction (TBVF) in the proximal femur and humerus in four felids that vary in their wild home ranges and locomotor/prey capture behaviours. This is a novel study. Variation in TBVF has not been studied before in captive vs. wild conspecifics and there are few studies of how morphology varies between captive/wild contexts in the postcranial skeleton in general. Introduction is clear and concise. Hypotheses are clearly articulated. The results are clearly presented and discussed. The results are interpreted with caution and the conclusions are well supported. There are impressively few typos. Overall, there is very little I can critique about this study or the manuscript; it was a joy to read and the results are exciting! The results are relevant to a broad range of disciplines, ranging from those interested in morphology/bone functional adaptation to animal conservation and thus appropriate for RS Open Science.

I list a few minor comments below. My only ‘major’ suggestion is this: I am struck by the differences in the degree of variation across some taxa between wild vs. captive values and between the femur and humerus. For example, the highly variable jaguar captive femur vs the wild femur, while the opposite is true in the mountain lions. It might be interesting to explore this variation in more detail (e.g. coefficient of variation) or to just discuss it in the Discussion. I would personally expect captive specimens might be more variable because enclosures, diet, activity levels, etc. would likely differ substantially across zoos/time period while, although wild animals might vary in frequencies of locomotion, terrain, etc., I would think their TBVF might be more constrained due to a more consistent/natural diet, activity levels, etc. However, this does not necessarily seem to be the case, which is quite interesting.

We thank the Reviewer for raising this point. We have now included a new paragraph in the Discussion that specifically addresses this issue, including presentation of coefficients of variation of BVF in captive and wild samples. As we note, it is possible that there would be greater variation in captive than wild animals, as suggested by the Reviewer, due to variation in captive living conditions; it is also possible that less variation in the captive species would result from an overall reduction in mobility. Our comparison of CVs shows no consistent patterning and great overlap between wild and captive animals, possibly as a result of these conflicting influences.

Minor comments:

Why use abbreviation ‘TBVF’ when the vast majority of studies use BV/TV and it seems the two

are considered equivalent here (pg 4., line 24: “resulting in a ratio known as bone volume/total volume (BV/TV), i.e., trabecular bone volume fraction, TBVF.” Is this because the pQCT bone mineral density values need to be converted to BV/TV values and the authors wish to make this distinction clear? If not, I would suggest using ‘BV/TV’ so it is clear that this study is analysing the same information as that other trabecular studies (including the Doube et al. 2011 study the authors cite in relation to the lack of significant allometric scaling).

Thank you for the comment. This is a point that we have grappled with previously in comparing results from 3D images derived from microCT and those from a pQCT. Although we have successfully (Chirchir et al. 2015) found a correlation between the two measures, we thought it would still be important to use a term that shows that we are not strictly referring to BV/TV only, but also data from a pQCT scanner that have previously been identified as trabecular bone volume fraction (or just trabecular fraction). If we were to use BV/TV, it is still referring to bone volume fraction. Therefore, for consistency, we use BVF to refer to bone volume fraction derived from both microCT and pQCT data.

pg 4/line 20: Why is the VOI scaled at 65-75% and not one standard scaling (e.g. all at 65%)? Does 65-75% mean it was 65% in some species and 75% in others, or 65-75% across all individuals depending on individual variation in the internal/external morphology?.

We tried to select similar VOI sizes for both pQCT- and microCT-derived data. We scaled within a range 65-75% because of the variation in size of the different species and elements. So, for each species with similar epiphyseal size we selected the same size VOI; e.g., for all leopards in the humerus we used 65% VOI. For all jaguars we used a VOI of 75% in the femur. We have clarified this point in the text.

Despite lack of allometric scaling in TBVF/BVTV, it would be helpful to know the rough estimates of body mass in each of the four felid species (either wild, captive or both if known) just so the reader has a better sense of the variation in body size across these four felids. Although I think most readers have a general sense of what these animals look like, it's harder to fully grasp their variation in body mass.

We have provided male and female average body mass for each species in Table 1.

pg 6, line 26: might be worth clarifying here that this was only in the femur. The leopard humerus was significantly different.

Corrected

Figure 1. Is it possible to highlight the VOI in a different colour (e.g. yellow) as the red is difficult to see against the black background

Corrected

Table 1. remove ‘all’ from the mountain lion, femoral head column ‘all pQCT scanned’ (or add ‘all’ to the other columns/species for consistency)

Corrected

Tracy Kivell

Appendix C

Dear Editor,

Thank you for the opportunity to make final revisions on this manuscript. We have provided our response to the minor revisions below.

Let us know if you have any questions about this.

Best,

Habiba on behalf of all co-authors

Associate Editor Comments to Author (Dr Marco Palanca):

Associate Editor: 1

Comments to the Author:

Please, add a statement about data availability for the scientific community: if data sharing is possible or if data are copyright of the museums.

We added the statement below.

Data accessibility: Femoral and humeral BVF data are provided in supplementary materials. These data were obtained from 3D images from microCT scanning and from single slice images from pQCT scanning. All images are available for sharing from the PI.

Reviewer comments to Author:

Reviewer: 2

Comments to the Author(s)

The authors have done an excellent job of addressing my comments, which were relatively minor, and those of the other reviewer. This was a strong paper to begin with and the changes that have been implemented in the revised version have made it even stronger. I am happy for this manuscript to be accepted and look forward to seeing it published.

Tracy Kivell

Reviewer: 3

Comments to the Author(s)

GENERAL COMMENTS

This paper investigates the effects of loading activities on bone remodeling by comparing felids in captivity and wild. The research question is clear and highly relevant. The study is nicely designed to address the research question. The methods are robust and the data are analyzed convincingly. I only have minor comments on this paper.

SPECIFIC COMMENTS

The paper is generally well-written and the images are clear. Here are some detailed comments:

1. INTRODUCTION, page 1, line 33: a closed bracket is missing

Corrected

2. INTRODUCTION, page 2, line 31 and following: I wonder whether hypothesis (iii) was really formulated before initiating the study, or was it more general, e.g. you originally suspected “only” that differences might exist between bones

We included this hypothesis due previous work showing differences in how fore and hindlimbs of mammals are used especially during locomotion. Being a study on functional morphology, it was necessary and important to address differences between bones. This is further explained in our justification of the hypothesis.

3. MATERIALS AND METHODS, PAGE 3: the information under the sub-heading “Locomotor behaviour in the wild” looks more like Introduction (or relevant to the discussion) than M&M
The Reviewer is correct in noting that the information provided could fit in the Introduction or Discussion however, the reason we provide this information in the Materials & Methods is because it falls within the sample (Materials) description. The goal of having that section there, is to provide the reader with information as to why we selected the four felid species. In this final revision, we have modified that subheading to read “**Locomotor behaviour in the wild of study samples**” to indicate that we are describing behavior of our study samples to justify our choice of samples.

4. TABLE 3: the P-values are definitely relevant. Not so sure about the F-values

We have deleted F-values